# Microstructural Features and Surface Hardening of Ultrafine-Grained Ti-6Al-4V Alloy through Plasma Electrolytic Polishing and Nitrogen Ion Implantation

Marina K. Smyslova, Roman R. Valiev [ID], Anatoliy M. Smyslov, Iuliia M. Modina [ID], Vil D. Sitdikov and Irina P. Semenova *[ID]

Institute of Physics of Advanced Materials, Ufa State Aviation Technical University, 12 K. Marx st., 450008 Ufa, Russia; smyslovam@yandex.ru (M.K.S.); rovaliev@gmail.com (R.R.V.); smyslovmail@gmail.com (A.M.S.); modina_yulia@mail.ru (I.M.M.); svil@ugatu.su (V.D.S.)
* Correspondence: semenova-ip@mail.ru; Tel.: +7-347-273-3422

**Abstract:** This work studies a near-surface layer microstructure in Ti-6Al-4V alloy samples subjected to plasma electrolytic polishing (PEP) and subsequent high-energy ion implantation with nitrogen (II). Samples with a conventional coarse-grained (CG) structure with an average α-phase size of 8 μm and an ultrafine-grained (UFG) structure (α-phase size up to 0.35 μm) produced by equal channel angular pressing were used in the studies. Features of phase composition and substructure in the thin surface layers are shown after sequential processing by PEP and II of both substrates with CG and UFG structures. Irrespective of a substrate structure, the so-called "long-range effect" was observed, which manifested itself in enhanced microhardness to a depth of surface layer up to 40 μm, exceeding the penetration distance of an implanted ion he. The effect of a UFG structure on depth and degree of surface hardening after PEP and ion-implantation is discussed.

**Keywords:** titanium alloys; ultra-fine grained structure; plasma electrolytic polishing; ion implantation; modified layer; substructure; microhardness; long-range effect



## 1. Introduction

In recent decades, large scientific capacity has been accumulated in the studies on the surface modification of structural materials, including steels and titanium alloys, to improve their service performance. Among the most widely used methods are plasma electrolytic polishing (PEP) and methods associated with the impact of concentrated energy flows, which include ion implantation (II) [1–5]. Parts treated with PEP have a surface roughness Ra of up to 0.01 μm, while microscopic marks and a defective layer with foreign inclusions are removed off the surface. A high quality of a polished surface improves the corrosion resistance and fatigue strength of materials [4]. The physics of ion implantation consists of the introduction of an alloying element into the surface layer of a part as a result of its bombardment with high kinetic energy ions [2,3]. The so-called "long-range effect" occurs, one of the manifestations of which is the formation of a complex dislocation substructure at a depth that is tens of times greater than the penetration depth of a doping ion [6]. The surface modification in conventional coarse-grained (CG) Ti-6Al-4V alloy by nitrogen ion implantation improves its tribological [7] and corrosion properties [5] and its creep rupture strength at 600 °C [8].

In recent years, ultrafine-grained (UFG) metals and alloys processed by severe plastic deformation techniques have attracted great interest [9–12]. Formation of a bulk UFG structure in titanium alloys makes it possible to significantly improve their service and technological properties (strength, fatigue resistance, superplasticity, etc.) [12]. Integration of these technological methods associated with the creation of a UFG state in the material volume and surface modification to improve performance of titanium alloys is an urgent task for modern mechanical engineering.

At the same time, there arise a number of unsolved problems connected with the effect of these modification methods on the surface of a substrate with a UFG structure. Since oxidation processes in UFG titanium alloys occur much more intensively, and thin but rather dense oxide films form, this can, for example, complicate the electrochemical process during PEP and, as a result, affect the physical and chemical state of the surface. During high-energy ion implantation, a complex dislocation substructure is formed in the surface layer, and ions embedded in the crystal lattice of a metal substrate form solid solutions or new chemical compounds [2]. In contrast to conventional CG materials, nucleation and accumulation of new dislocations in ultrafine grains with high dislocation density are hindered [11]. This can probably affect the mechanisms of modified layer formation in the surface of UFG titanium alloy and, therefore, its service characteristics.

The work is aimed at studying the effect of a Ti-6Al-4V substrate UFG structure on structural and phase changes in the alloy surface layers after plasma electrolytic polishing and ion implantation with nitrogen.

## 2. Experimental Procedure

### 2.1. Materials and Sample Preparation Procedure

The investigations were carried out on Ti-6Al-4V alloy. The chemical composition is presented in Table 1.

**Table 1.** Chemical composition of Ti-6Al-4V alloy (in wt.%).

| Ti | Fe | C | Al | O | V | N | H | Si | Zr |
|-------|------|-------|-----|------|-----|------|-------|-------|------|
| basis | 0.18 | 0.007 | 6.6 | 0.17 | 4.9 | 0.01 | 0.002 | 0.033 | 0.02 |

Equal channel angular pressing (ECAP) was used to form the UFG structure in the Ti-6Al-4V alloy. The microstructure of as-received billets was predominantly equi-axed with an average grain size of the $\alpha$-phase of 15 μm, which is typical of hot-rolled rods. Rods with a diameter of 20 mm and 100 mm in length were preliminarily heat-treated to produce a mixed duplex (globular–lamellar) microstructure according to the following regime: quenching at T = 960 °C (by ~30 °C below the $\beta$-transus temperature) for 1 h, followed by tempering at T = 675 °C for 4 h. Then, the rods were deformed in a die-set with an angle of the channels intersection at 120° at T = 650 °C along the Bc route, 6 passes with a total accumulated strain $\varepsilon$~3 [13].

Disc-shaped samples with a thickness of 5 mm were cut out of the rods (Figure 1a). Before ion implantation, the sample surface was subjected to mechanical polishing and plasma electrolytic polishing to a mirror finish. The details of PEP processing are given in [14]. The ion implantation of the sample surface was carried out on an experimental set with a broadband ion source. Before implantation, the samples were washed with gasoline in an ultrasound bath and wiped with ethanol. Based on the results of earlier studies [15] on Ti-6Al-4V alloy (Russian analog VT6), the following treatment modes were selected: the dose of ion implantation $D = 2 \times 10^{17}$ ion/cm$^2$; the accelerating voltage $U_{acc}$ = 25 kV; the residual pressure in the chamber $p$ = 5.5 . . . 6.5 Pa; the heating temperature of samples in the chamber did not exceed T = 250 °C. After implantation the samples were cooled in vacuum for 2–3 min.

### 2.2. Structure Studies in the Bulk and Surface Layer of Samples

The microstructure was studied by scanning electron microscopy (SEM) using a JEOL JSM 6390 microscope (JEOL Ltd., Tokyo, Japan) and transmission electron microscopy (TEM, JEOL Ltd., Tokyo, Japan) on a JEOL JEM 2100 with an accelerating voltage of 200 kV. Thin foils for TEM were prepared by spark cutting plates with a thickness of 0.8 mm in accordance with the scheme displayed in Figure 1a. Then, the foils were mechanically thinned till a thickness of ~100 μm and electro-polished using a TenuPol-5 facility with a solution of 5% perchloric acid, 35% butanol, and 60% methanol at a polishing temperature

in the range from −20 to −35 °C. Only the non-polished side of the foils was thinned (Figure 1b). The foils allowed for revealing structural changes in the near-surface layer at a depth of no more than 200 nm.

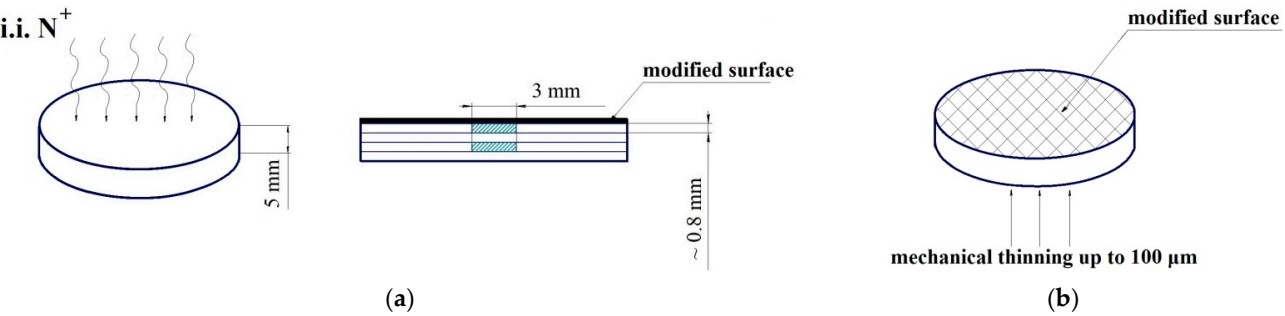

**Figure 1.** Scheme of sample processing and preparation for the studies: disc-shaped samples for plasma electrolytic polishing and ion implantation (**a**); scheme of mechanical thinning of foils for microstructure studies by TEM (**b**).

### 2.3. X-ray Structure Analysis

The study on the surface layer structural-phase and chemical composition was carried out by X-ray diffraction analysis (XRD). To determine the size of coherent scattering domains (CSD) and dislocation density ($\rho$), X-ray diffraction patterns were obtained on a Rigaku Ultima IV diffractometer (Rigaku Corporation, Tokyo, Japan), which implements a focusing method (Bragg–Brentano goniometer scheme). All X-ray diffraction patterns were obtained using Cu$K\alpha$ radiation at a voltage and current of 40 kV and 40 mA, respectively, on the X-ray tube. The diffraction patterns were measured within the scattering angle $2\theta$ from 30° to 150° with a step of 0.02° and exposure of 4 s per point. A graphite monochromator (Rigaku Corporation, Tokyo, Japan) was used on a reflected beam. The WPPM approach implemented in the PM2K software (version 2.1, University of Trento, Trento, Italy) was used to analyze the diffraction patterns in order to calculate the distribution of CSD over size, density of edge and screw dislocations, and effective radius of dislocations [16,17]. Such parameters as sample plane displacement, lattice parameter a, dislocation density $\rho$, volume fraction of edge dislocations $m_{ixp}$, effective dislocation radius $R_e$, and the shape and size of crystallites $D$ were varied to calculate diffraction patterns according to [16,17]. The average crystallite size $D_{ave}$ was set in the form of a sphere with lognormal distribution $D_{ave} = \exp(\mu + \sigma^2/2)$, where $\mu$ and $\sigma$ are the average value and its deviation, respectively. The instrumental broadening of diffraction lines, i.e., the parameters $U$, $V$, $W$, $a$, $b$, and $c$ of the Cagliotti function were determined by processing the X-ray diffraction patterns of LaB$_6$ obtained under the same conditions in which the titanium alloy was studied. The diffraction patterns were calculated as a result of 40 iterations with the initial parameters equal to the following: $a = 0.2956$ nm, $c = 0.4687$ nm, $\rho = 1.0 \times 10^{14}$ m$^{-2}$, $m_{ixp} = 0.5$, $R_e = 3.0$ nm, $\mu = 4.0$, $\sigma = 0.1$, $W = 1.93000000 \times 10^3$, $V = 6.27346000 \times 10^4$, $U = 2.03000000 \times 10^3$, $a = 2.38030000 \times 10^1$, $b = 9.93000000 \times 10^3$, and $c = 0$.

The qualitative phase analysis was performed using a PDF-2 X-ray database in the PDXL software package (version 1.8.1, Rigaku Corporation, Tokyo, Japan). To enhance the quantitative phase analysis accuracy, the volume fraction $f$ of precipitates was calculated by the Rietveld method [18]; the pseudo-Voigt function was applied to describe the shape of a peak profile, with account of the asymmetry of a peak, subtracting the background radiation by the Sonneveld–Visser method [19]. All this made it possible to calculate precipitates with a volume fraction of less than 0.5%.

### 2.4. Microhardness

The surface microhardness was measured using a Struers Duramin microhardness tester (Struers A/S, Ballerup, Denmark) with a load of 25 g for 10 s. The microhardness was measured on the surface of the studied samples, as well as on the inclined microspecimens

prepared according to the scheme shown in Figure 2. The investigated surface was inclined to the initial one at an angle of about 2°. At least 5 measurements were performed at each point at depth.

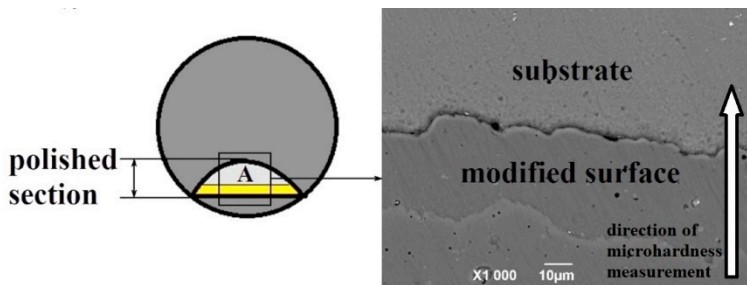

**Figure 2.** Scheme of microhardness measurement over the sample depth.

### 3. Results

*3.1. Microstructure of VT6 in CG and UFG States*

The initial microstructure of a heat-treated workpiece was a mixed globular–lamellar structure (Figure 3a) and consisted of a primary α-phase (the average size was about 5 μm, the fraction was about 30%) and areas with (α + β) thin-lamellar structure (Figure 3b). The thickness of plates was no more than 1 μm on average (Figure 3b). This type of structure ensures good technological ductility of the material and the most effective refinement of the alloy under severe plastic deformation by ECAP. This approach was described earlier in [20]. The results of the X-ray phase analysis revealed that the ratio of the α and β phases was approximately 85:15%.

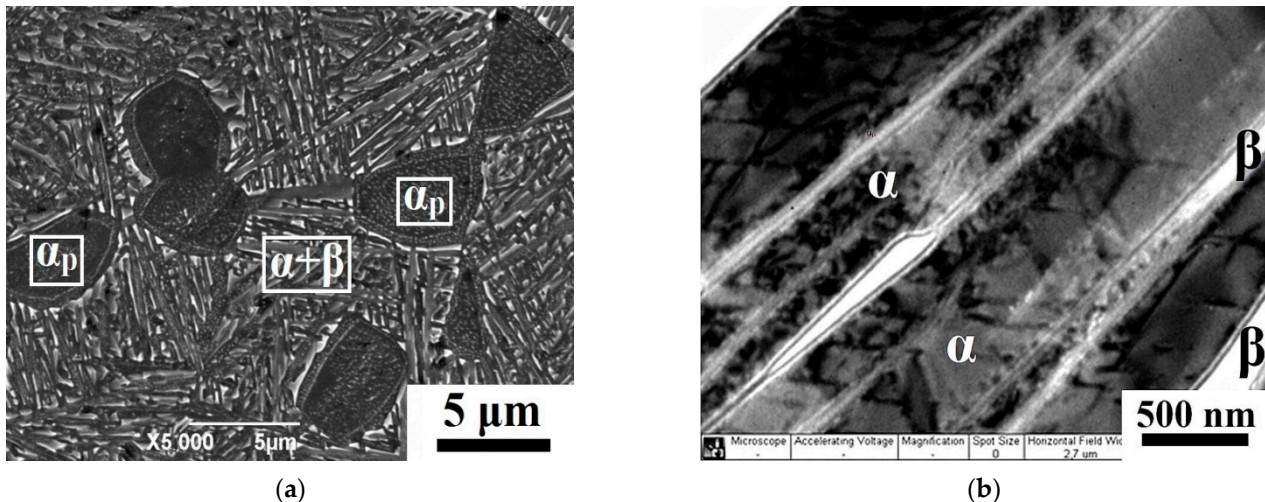

(**a**)          (**b**)

**Figure 3.** Microstructure of Ti-6Al-4V in the initial CG state: general view of structure, SEM (**a**); (α + β) area of lamellar structure, TEM (**b**).

Figure 4 shows the alloy microstructure after ECAP obtained by SEM and TEM. Weakly deformed grains of a primary α-phase with an average size of about 5 μm were observed in the microstructure. In the two-phase α + β areas, a mixture of α- and β-phases in the form of ultrafine grains was found (Figure 4a). The TEM images show that the average grain/subgrain size was about 350 nm (Figure 4b). The interiors of the α-grains were characterized by a high density of dislocations and a cellular-type dislocation substructure formed inside the weakly deformed primary α-phase (card No. 00-044-1294). The fraction of the β-phase, identified with card No. 00-044-1288, decreased from 15 to 6%, compared to the initial state, as a result of partial decomposition and dissolution $\beta_m \rightarrow \alpha + \beta$ initiated by severe plastic deformation [21]. The total dislocation density

in the workpiece after SPD, as estimated by X-ray analysis, was about $12 \times 10^{14}$ m$^{-2}$, which was significantly higher than the dislocation density in the initial state of the alloy ($0.05 \times 10^{14}$ m$^{-2}$) (see Table 2). The microhardness of the sample surface before and after ECAP was 3800 and 4500 MPa, respectively.

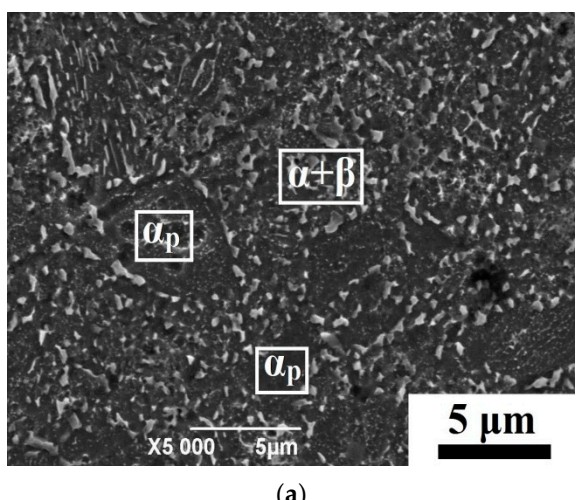

(**a**)

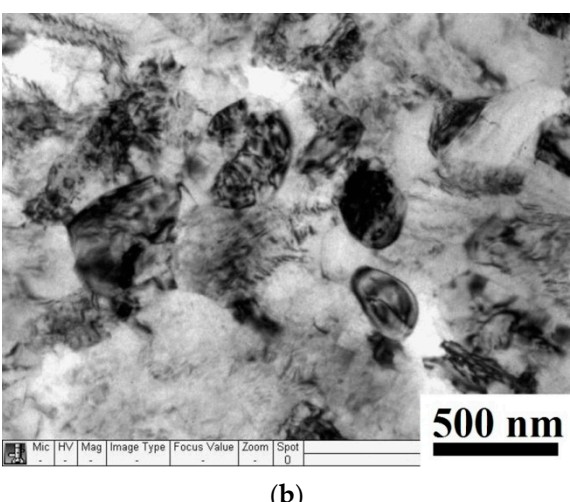

(**b**)

**Figure 4.** Microstructure of Ti-6Al-4V alloy in the UFG state: general view of the structure, SEM (**a**); (α + β) area of a mixture of ultrafine grains, TEM (**b**).

**Table 2.** Results of X-ray analysis.

| No. | Surface Treatment | CSD, nm | Dislocations Density, $\rho \times 10^{14}$ m$^{-2}$ | β-Ti, % | TiN, % | TiO2, % |
|---|---|---|---|---|---|---|
| 1 | CG annealed, no treatment | 93 | 0.05 | 15.0 | - | - |
| 2 | CG + PEP | - | 2.0 | - | - | - |
| 3 | CG + PEP + II | 42 | 10.0 | 15.0 | 0.12 | 0.12 |
| 4 | UFG no treatment | 23 | 12.0 | 8.5 | - | - |
| 5 | UFG + PEP | - | 9.0 | - | - | - |
| 6 | UFG + PEP + II | 63 | 23.0 | 7.5 | 0.10 | 0.10 |

### 3.2. Microstructure of the Near-Surface Layer of Samples after PEP

The near-surface layer microstructure in the CG and UFG sample states after PEP treatment is shown in Figures 5 and 6.

The TEM images showed the formation of a reticular and inhomogeneously cellular dislocation substructure in the near-surface layer in coarse grains of a primary α-phase (Figure 5a). Dislocation pile-ups were observed at the interphase boundaries of primary α-phase grains and between plates (Figure 5b). This is consistent with the results of the X-ray diffraction analysis of the CG samples surface after PEP, which noted a significant increase in the dislocation density from 0.05 to $2.0 \times 10^{14}$ m$^{-2}$ (see Table 2). Microdischarges arising in the PEP process probably promote the generation of new dislocations, which accumulate on obstacles in the form of interphase boundaries.

The investigation of UFG samples after PEP revealed that the microstructure retained its pattern in the sample surface, but it became more inhomogeneous due to the redistribution and partial annihilation of dislocations inside and along the boundaries of ultrafine grains/subgrains (Figure 6a). The largest pile-ups of dislocations were observed in the grain and phase boundaries (Figure 6b). The TEM data agree with the X-ray analysis results with respect to the total dislocation density decreasing slightly from 12.0 to $9.0 \times 10^{14}$ m$^{-2}$, in contrast to the CG substrate (see Table 2). The total value of dislocation density in the

near-surface zone of the UFG substrate remains several times higher than that of the CG alloy (9.0 and $2.0 \times 10^{14}$ m$^{-2}$, respectively).

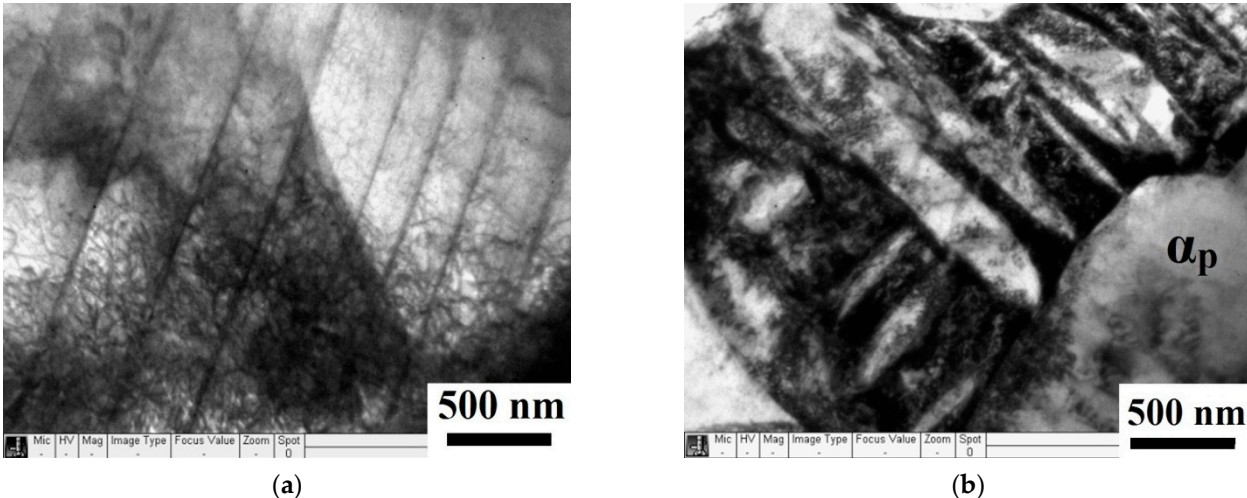

(**a**)  (**b**)

**Figure 5.** Microstructure of the near-surface layer of a Ti-6Al-4V sample with CG structure after PEP: dislocation substructure in the primary $\alpha_p$ (**a**); ($\alpha + \beta$) lamellar structure (**b**). TEM.

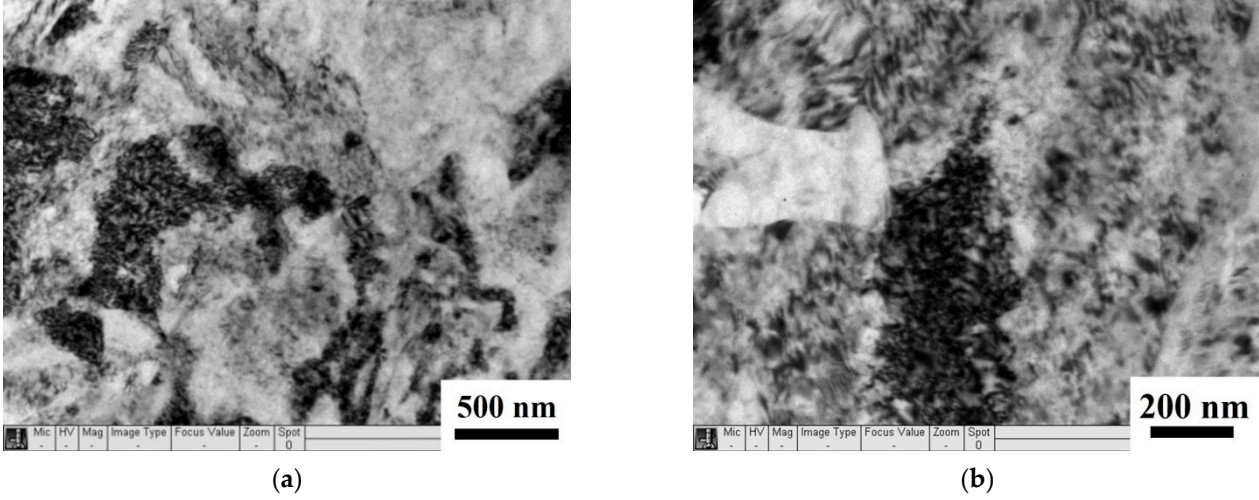

(**a**)  (**b**)

**Figure 6.** Microstructure of the near-surface layer of a Ti-6Al-4V sample with UFG structure after PEP: general view of the inhomogeneous structure (**a**); pile-ups of dislocations (**b**). TEM.

### 3.3. Microstructure of a Sample Modified Layer after Ion Implantation

Figures 7 and 8 show TEM images of the near-surface layer of the CG and UFG substrates of Ti-6Al-4V alloy after ion implantation with nitrogen. By comparing the layer microstructure after PEP (Section 3.1) and II, one can note that, due to the impact of ions with high kinetic energy, the structure in both CG and UFG substrates undergoes significant changes, which leads to the formation of a strong state of nonequilibrium with a large number of crystal defects as a result of radiation-stimulated diffusion [3].

In the CG substrate, strong disordering of the structure (Figure 7a) and the formation of blocks with high dislocation density are observed (Figure 7b). According to the results of X-ray structural analysis, a noticeable decrease in the CSD size (from 93 to 42 nm) was found, and the total dislocation density increased by another five times: from 2 to $10 \times 10^{14}$ m$^{-2}$ compared to the surface state after PEP (see Table 2).

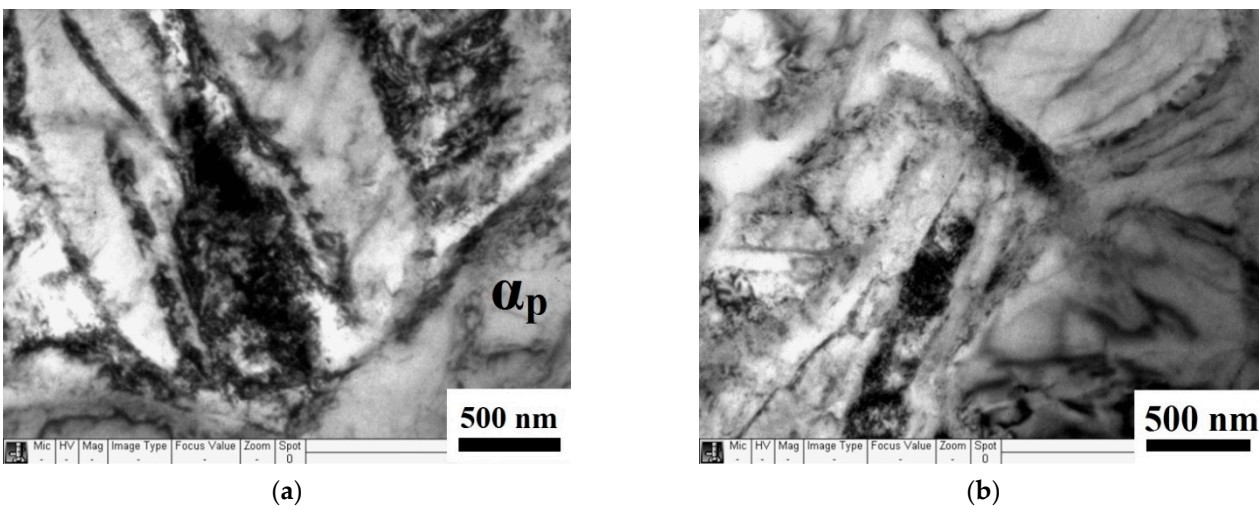

**Figure 7.** Microstructure of a near-surface layer of a CG Ti-6Al-4V sample after PEP + II: general view of structure disordering (**a**); blocks with high dislocation density (**b**). TEM.

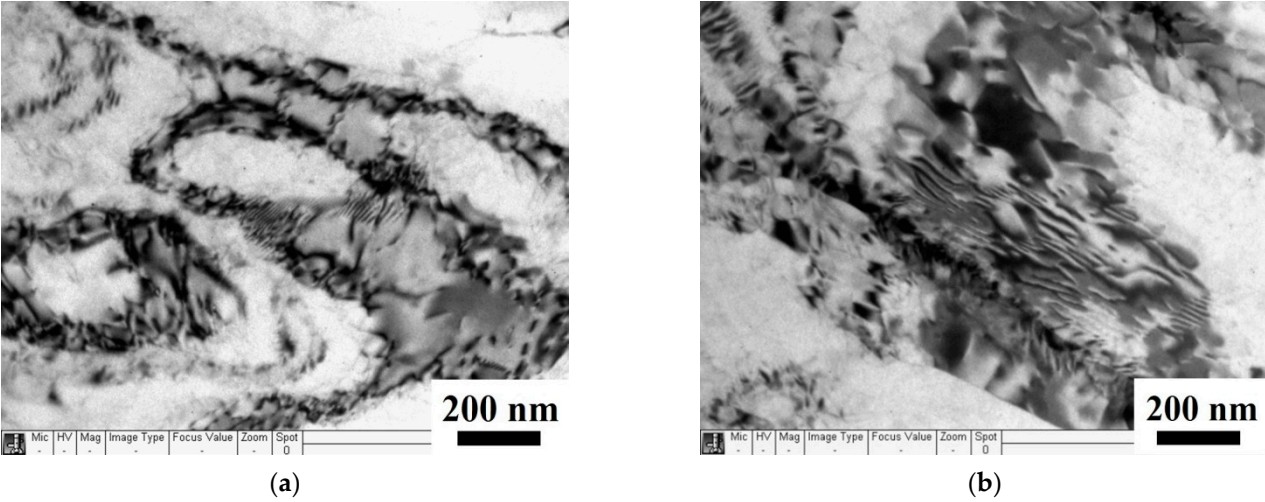

**Figure 8.** Microstructure of the near-surface layer of a Ti-6Al-4V sample with UFG structure after PEP + II: accumulations of crystal defects along subgrain boundaries (**a**); redistribution and partial annihilation of dislocations (**b**). TEM.

In the UFG substrate, accumulations of dislocations and other crystal defects were also observed, mainly along subgrain boundaries (Figure 8a). In separate areas (as well as in subgrains), one can observe a slight decrease in the dislocation density as a result of their redistribution and partial annihilation (Figure 8b). The dislocation density grew only by 2.5 times in comparison with a non-modified state (from 9.0 to $23.0 \times 10^{14}$ m$^{-2}$), in contrast to the CG substrate, the dislocation density of which increased by five times after ion implantation (see Table 2). Apparently, after severe plastic deformation, the dislocation density had already reached critical values; therefore, ion bombardment of a UFG substrate surface led to their partial annihilation. This is also evidenced by an increase in the CSD size to 63 nm (see Table 2). The dislocation density in the ion-implanted surface of a UFG sample remains much higher than that in the surface of a CG alloy after ion implantation (23 and $10 \times 10^{14}$ m$^{-2}$, respectively). This can be explained by the pinning of dislocations on nitride and oxide precipitates as a result of nitrogen ions' penetration into the crystal lattice [22]. This is confirmed by the results of X-ray structural analysis in Table 2. The volume fraction of nitride (card No. 00-038-1420) and oxide (card No. 00-021-1272) precipitates in both cases is approximately the same, and is in the range of 0.10–0.12%.

It should be noted that ion implantation with nitrogen did not lead to significant changes in the ratio of α- and β-phases in CG and UFG substrates (see Table 2).

### 3.4. Microhardness of a Sample Modified Layer after Ion Implantation

Figure 9 displays the results of microhardness measurements in the CG and UFG substrates subjected to implantation with nitrogen. The microhardness was measured on the inclined samples in accordance with Figure 2. One can see that the maximum value of microhardness in both substrates is approximately the same, and amounts to about 5300 MPa. At a depth of ~5 μm, the microhardness slightly decreases to 5000 MPa in UFG and to 4700 in CG samples. It should be noted that the earlier studies on an ion-implanted surface of a UFG alloy showed that the penetration depth of nitrogen ions did not exceed 6 μm [15].

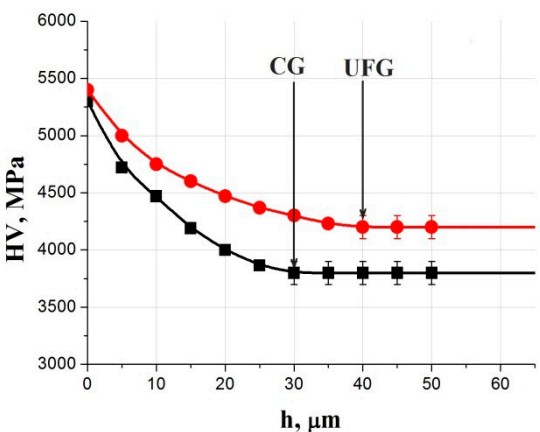

**Figure 9.** Microhardness behavior over the depth of UFG and CG substrates of a Ti-6Al-4V alloy after high-energy nitrogen ion implantation.

The microhardness in the surface layer is enhanced in comparison with that in the internal volume of samples in both CG and UFG states and retains practically up to a depth of 30 and 40 μm, respectively (Figure 9). The researchers attribute this to a "long-range effect" during ion implantation, when a change in the dislocation substructure in the surface was observed in deeper layers several times exceeding the implantation depth of ions [6,23].

## 4. Discussion

This work presented some features of a structural-phase state of a modified surface layer of samples made of ultrafine-grained Ti-6Al-4V alloy treated by plasma electrolytic polishing followed by ion implantation with nitrogen.

Plasma electrolytic polishing is usually performed to reduce the surface roughness of products. It is a preparatory operation for subsequent modification by ion implantation and/or the application of various coatings. As is known, the microstructure of a substrate affects the electrochemical activity of a surface. In particular, the transition of a metal to a UFG state, which is characterized by a high density of boundaries and defects, is accompanied by the enhanced activity of electrons and an increased diffusion coefficient in grain boundary areas [24]. The indicated effect of electrochemical activity enhancement can be explained by intensification in the heterogeneity of the surface substructure and its thermodynamically unstable state [25]. These differences in the electrochemical behavior of the CG and UFG alloys affected the dislocation substructure in the near-surface layers of the samples after treatment. First of all, PEP processing of the CG substrate surface led to its strengthening due to an increase in the total dislocation density from 0.05 to $2.0 \times 10^{14}$ m$^{-2}$. On the contrary, partial softening and dislocation density reduction from 12.0 to $9.0 \times 10^{14}$ m$^{-2}$ were observed in the UFG substrate, although its value is several

times higher than that of the CG alloy. The UFG material, unlike the CG one, already has a high degree of hardening and a state of less equilibrium. At energy pulses, probably from microdischarges during PEP, the generation of motion of unstable dislocation clusters occurs, leading to stress relaxation and some softening of the UFG sample surface. Thus, as applied to UFG titanium alloy, PEP can be an independent process for surface treatment, allowing one to achieve high surface quality characteristics that are not attainable by other processing methods [14].

Ion implantation is now widely used for steels and titanium alloys. It is known that the surface hardening of titanium alloys is achieved as a result of the interaction of implanted nitrogen ions with the resulting radiation defects of the crystal structure, as well as a significant solid solution hardening mechanism due to the formation of dispersed nitride phases [7,8]. A similar effect of nitrogen ion implantation in the surface was observed in both CG and UFG substrates. The total density of dislocations in the near-surface layer with CG structure increased by five times, to $10 \times 10^{14}$ m$^{-2}$, and that with UFG structure grew only by 2.5 times (Table 2). In both samples, strong disordering of the structure was observed because of large distortions of the crystal lattice as a result of shock-wave action during ion implantation and the formation of a large number of radiation defects. A significant increase in the dislocation density in the near-surface zone of the CG and UFG samples affected the microhardness value (Figure 9). Its values were 5000 and 4700 MPa for UFG and CG specimens, respectively. In this case, no significant difference in the microhardnesses of CG and UFG structured surfaces was observed. This is probably due to the UFG surface having a critical dislocation density before ion implantation. Therefore, further bombardment led to their redistribution and partial annihilation. This fact is proven by the different increment in the dislocation density after ion implantation as compared to the no treatment state of CG (from 0.02 to $10 \times 10^{14}$ m$^{-2}$) and UFG (from 12 to $23 \times 10^{14}$ m$^{-2}$) substrates (Table 2)

It should be noted that a high microhardness in the UFG sample was approximately at a depth of up to 5 μm, which is consistent with the depth of nitrogen ions penetrating into the surface, which was determined using the Auger spectroscopy in [15]. The same depth of nitrogen ion penetration was observed in the surface of conventional titanium alloy [26]. The inhibition of a doping ion in the near-surface layer of the UFG substrate probably occurs similarly to that in the CG state, i.e., the processes in thin layers of ion penetration are insensitive to the substrate structure. The formation of highly dispersed segregations and nitride phases is confirmed by X-ray diffraction analysis, according to the results of which, an average of 0.10–0.12% nitride phases was found in the irradiated surface of the CG and UFG substrates. A similar result was obtained in our recent work [27].

Enhanced microhardness, as compared to that in the material bulk, was found in the ion-implanted surface of both CG and UFG substrates. In the UFG material it retains at a depth of up to 40 μm, and in the CG alloy it was observed at a depth of 30 μm (Figure 9). Such an unusual effect was demonstrated in earlier studies devoted to the ion implantation of titanium alloys [26]. The authors previously identified a superdeep change in the structure of the implanted surface, which manifested itself in the formation of a complex dislocation structure at a depth of up to 30 μm. A model of deep formation of a developed dislocation structure was proposed. During ion bombardment of the surface, vacancy dislocation loops are formed in the zone of atomic collision cascades as a result of diffusion-induced rearrangement. The dislocation loops grow and transform under the conditions of increasing concentration of point defects (vacancies), which also contributes to the mass transfer of impurity atoms. An increase in the depth of a hardened surface layer in the UFG alloy up to 40 μm is apparently due to a more active redistribution of dislocation formations, the creation of new stable configurations, and the consolidation of nitride phase segregations within them.

The hardening of the UFG surface depends on treatment conditions (annealing before or after ion implantation, number of implantation cycles). It was demonstrated in [27] that a three-time iteration of the cycle (ii + annealing) resulted in the maximum concen-

tration of nitride phases (~0.28%). Subsequent low-temperature annealing (to 400 °C) after ion implantation can lead to the formation of more equilibrium and an ordered grain boundary structure, and, consequently, to the enhancement of both strength and ductile properties [27].

The investigations on various structural states of Ti-6Al-4V alloy have shown that a completely new structural state is formed in the substrate surface during sequential PEP and II processes. The new state is associated with an increase in the dislocation density and the creation of a complex dislocation substructure, which indicate successive hardening of the material at all the stages of processing. The UFG state, initially already having a high hardening degree and less equilibrium, is capable of generating a motion of unstable dislocation clusters, leading to relaxation of stresses, their equalization, and some softening with energy pulses from microdischarges during PEP. During subsequent nitrogen ion implantation of the surface of UFG samples, the depth of ion penetration is similar to that of the CG state. The long-range effect as a result of ion bombardment is significantly higher, the surface hardening is retained at a greater depth in comparison with the CG substrate. However, in order to achieve the desired effect of UFG substrate surface hardening, one should pay attention to the processing regimes, which can be varied, for example, by a number of repeated cycles of ion implantation, annealing conditions, etc. This will be the focus of our further investigations.

### 5. Conclusions

It was found that the initial structural state of the Ti-6Al-4V alloy substrate had a significant effect on the transformation of the dislocation substructure during PEP and subsequent ion implantation with nitrogen:

1. PEP leads to formation of new dislocation configurations and dislocation density enlargement by almost an order of magnitude in the surface of the CG alloy. In thin surface layers of the substrate with UFG structure, the dislocation pile-ups level up, accompanied by a slight decrease in the total density of dislocations;
2. Irrespective of the structural state in the substrate surface after the subsequent high-energy nitrogen ion implantation, a "long-range effect" was observed, which manifested itself in strengthening to a depth up to 30–40 μm;
3. Ion implantation with nitrogen leads to nitride and oxide precipitation with a volume fraction of 0.10–0.12% in the near-surface layers, which makes an additional contribution to surface hardening.

**Author Contributions:** Conceptualization and discussion of the results, M.K.S. and A.M.S.; investigation, V.D.S., I.M.M. and R.R.V.; collection of data and writing the draft, I.P.S. All authors have read and agreed to the published version of the manuscript.

**Funding:** This research was funded by the Russian Science Foundation, grant number 19-79-10108.

**Acknowledgments:** Authors are grateful to the personnel of the research and technology Joint Research Center, 'Nanotech', Ufa State Aviation Technical University for their assistance with instrumental analysis.

**Conflicts of Interest:** The authors declare no conflict of interest.

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
