# Peer review of "Microstructural Features and Surface Hardening of Ultrafine-Grained Ti-6Al-4V Alloy through Plasma Electrolytic Polishing and Nitrogen Ion Implantation"

_metals, doi:10.3390/met11050696_

Round 1
Reviewer 1 Report
The reviewed article deals with microstructure and surface hardening of UFG Ti-6Al-4V alloy modified by plasma electrolytic polishing and ion implantation. Authors presented interesting and deep investigations of the microstructural features. Nevertheless there are some minor remarks:
- The better way to show chemical composition is present it in table.
- XRD analysis - what was the range of 2teta angle, what were the step and time for step values during investigations?
- Please add ICDD data base for used diffraction patterns.
- How many measurements have been carried out for one depth?
- Why Authors used classical microhardness measurements instead instrumental indentation?
The most important remark - there is almost no effect of surface hardening when we compare hardness for CG and UFG surface. Please explain this phenomenon.
Additional, ion implantation did not have an important influence on hardening features. This should be also discussed.
Author Response
The reviewed article deals with microstructure and surface hardening of UFG Ti-6Al-4V alloy modified by plasma electrolytic polishing and ion implantation. Authors presented interesting and deep investigations of the microstructural features. Nevertheless there are some minor remarks:
- The better way to show chemical composition is present it in table.
Authors are agree. The chemical composition is presented in Table 1.
- XRD analysis - what was the range of 2teta angle, what were the step and time for step values during investigations?
Agree. This information is given in section 2.3.
- Please add ICDD data base for used diffraction patterns.
Agree. This information is given in section 3.
- How many measurements have been carried out for one depth?
The number of measurements was at least 5 per point in depth (see Section 2.4)
- Why Authors used classical microhardness measurements instead instrumental indentation?
In this work, we measured the microhardness at the minimum possible indenter load of 25 g. In our opinion, when comparing samples with UFG and coarse-grained (CG) structure, the classical approach allows obtaining results that are more objective. In the case of nanoindentation of the CG surface, in which the size of separate phases reaches several micrometers, the scatter of HV values will be significant.
The most important remark - there is almost no effect of surface hardening when we compare hardness for CG and UFG surface. Please explain this phenomenon. Additional, ion implantation did not have an important influence on hardening features. This should be also discussed.
Agree with this remark. We have supplemented Section 4 Discussion.
Authors are grateful to the reviewer for the attention given to our manuscript. All the corrections in the text and references are highlighted in yellow.

Reviewer 2 Report
The paper is relevant and interesting for scienific community.
Please, check the lines 47-48 (thin/thick films) and the line 48 (PEP instead of EPP).
Please, use the term "ethanol" instead of "spirit" (line 74).
Author Response
The paper is relevant and interesting for scientific community.
Please, check the lines 47-48 (thin/thick films) and the line 48 (PEP instead of EPP).
Ок. In the text, the abbreviation “EPP” has been replaced by “PEP”.
Please, use the term "ethanol" instead of "spirit" (line 74).
Ок. In the text, the term “spirit” has been replaced by “ethanol”.
Authors are grateful to the reviewer for the attention given to our manuscript. All the corrections in the text and references are highlighted in yellow.

Reviewer 3 Report
While this work focused on microstructural features and surface hardening of ultrafine-grained Ti-6Al-4V alloy through plasma electrolytic polishing and nitrogen ion implantation, it still remains to be improved. From my point of view, I recommend this manuscript to be accepted for publication after a relatively major revision. Some suggestions for the authors to improve the manuscript, therefore, are as follows:
- Please check out all images carefully to keep the scale same. For example: the scale of figure 3 (a) and (b).
- Withall due respect, the annotation should be described clearly. Adding (a) and (b) in the figure 6-8.
- Please adjust the format of references to keep same.
- Please explain how the globular-lamellar structure forms.
Author Response
While this work focused on microstructural features and surface hardening of ultrafine-grained Ti-6Al-4V alloy through plasma electrolytic polishing and nitrogen ion implantation, it still remains to be improved. From my point of view, I recommend this manuscript to be accepted for publication after a relatively major revision. Some suggestions for the authors to improve the manuscript, therefore, are as follows:
- Please check out all images carefully to keep the scale same. For example: the scale of figure 3 (a) and (b).
We brought all the scales of the images to close values.
However, Figure 3 (a) was obtained using a scanning electron microscopy (SEM), and (b) using a transmission electron microscopy (TEM).
- With all due respect, the annotation should be described clearly. Adding (a) and (b) in the figure 6-8.
Agree with the comment of the reviewer. Changes have been made in captions to Figures 6-8.
- Please adjust the format of references to keep same.
The references were edited to the same format.
- Please explain how the globular-lamellar structure forms.
The microstructure of as-received billets was predominantly equi-axed with an average grain size of α-phase of 15 micrometer, which is typical of hot-rolled rods. Rods with a diameter of 20 mm and 100 mm in length were preliminarily heat-treated to produce a mixed duplex (globular-lamellar) microstructure (see Section 2.3).
Authors are grateful to the reviewer for the attention given to our manuscript. All the corrections in the text and references are highlighted in yellow.

Round 2
Reviewer 1 Report
All my remarks have been included.